# Blood vessel occlusion by *Cryptococcus neoformans* is a mechanism for haemorrhagic dissemination of infection

Josie F. Gibson[1,2], Aleksandra Bojarczuk[1,3]⊙, Robert J. Evans[1]⊙, Alfred Alinafe Kamuyango[1], Richard Hotham[1], Anne K. Lagendijk[4], Benjamin M. Hogan[4], Philip W. Ingham[2,5], Stephen A. Renshaw[1], Simon A. Johnston[1] *

1 Department of Infection, Immunity and Cardiovascular disease, Bateson Centre and Florey Institute, University of Sheffield, United Kingdom, 2 Institute of Molecular and Cell Biology, Agency of Science, Technology and Research (A-Star), Singapore, 3 Faculty of Physical Education, Gdansk University of Physical Education and Sport, Gdansk, Poland, 4 Division of Genomics of Development and Disease, Institute for Molecular Bioscience, University of Queensland, Brisbane, Australia, 5 Lee Kong Chian School of Medicine, Nanyang Technological University, Singapore

⊙ These authors contributed equally to this work.
* s.a.johnston@sheffield.ac.uk

**Data Availability Statement:** All relevant data are within the manuscript and its Supporting Information files.

## Abstract

Meningitis caused by infectious pathogens is associated with vessel damage and infarct formation, however the physiological cause is often unknown. *Cryptococcus neoformans* is a human fungal pathogen and causative agent of cryptococcal meningitis, where vascular events are observed in up to 30% of patients, predominantly in severe infection. Therefore, we aimed to investigate how infection may lead to vessel damage and associated pathogen dissemination using a zebrafish model that permitted noninvasive *in vivo* imaging. We find that cryptococcal cells become trapped within the vasculature (dependent on their size) and proliferate there resulting in vasodilation. Localised cryptococcal growth, originating from a small number of cryptococcal cells in the vasculature was associated with sites of dissemination and simultaneously with loss of blood vessel integrity. Using a cell-cell junction tension reporter we identified dissemination from intact blood vessels and where vessel rupture occurred. Finally, we manipulated blood vessel tension via cell junctions and found increased tension resulted in increased dissemination. Our data suggest that global vascular vasodilation occurs following infection, resulting in increased vessel tension which subsequently increases dissemination events, representing a positive feedback loop. Thus, we identify a mechanism for blood vessel damage during cryptococcal infection that may represent a cause of vascular damage and cortical infarction during cryptococcal meningitis.

## Author summary

Meningitis is a life threatening form of infection in the brain that is difficult to treat. How infection spreads from the blood to cause meningitis is not well understood. Here we have shown how infection with the fungus *Cryptococcus neoformans* can be spread from the

**Funding:** JFG was supported by an award from the Singapore A*STAR Research Attachment Programme (ARAP) in partnership with the University of Sheffield. Work in the PWI lab was funded by the A*STAR Institute of Molecular and Cell Biology (IMCB) and the Lee Kong Chian School of Medicine. RJE was supported by a British Infection Association postdoctoral fellowship (https://www.britishinfection.org/). AKL was supported by a University of Queensland Postdoctoral Fellowship. BMH by an NHMRC/National Heart Foundation Career Development Fellowship (1083811). SAJ, AB, RJE, AK and RH, were supported by Medical Research Council and Department for International Development Career Development Award Fellowship MR/J009156/1 (http://www.mrc.ac.uk/). SAJ was additionally supported by a Krebs Institute Fellowship (http://krebsinstitute.group.shef.ac.uk/), and Medical Research Council Centre grant (G0700091). AK was supported by a Wellcome Trust Strategic Award in Medical Mycology and Fungal Immunology (097377/Z/11/Z). SAR was supported by a Medical Research Council Programme Grant (MR/M004864/1). Light sheet microscopy was carried out in the Wolfson Light Microscopy Facility, supported by a BBSRC ALERT14 award for light-sheet microscopy (BB/M012522/1). Funders had no role in the study design, data collection and analysis, decision to publish, or preparation of the manuscript.

**Competing interests:** The authors have declared that no competing interests exist.

blood by blocking and bursting blood vessels. Using zebrafish larvae, we were able to follow the same infections over a period of days to understand how this infection behaves in blood vessels. We found that fungal cells become stuck within blood vessels depending on their size. These cells grow within blood vessels, resulting in the blood vessels becoming wider. We measured increased tension in blood vessels suggesting that, with the blockage and widening of vessels, there was increased local blood pressure. We found that vessel blockage was associated with their rupture and spreading of fungus into the surround tissue. Finally, by increasing the tension in vessels we could increase the number of blood bursting events supporting our conclusion that blood vessel blockage leads to the spread of the infection outside of blood vessels.

## Introduction

Life threatening systemic infection commonly results from tissue invasion following dissemination of microbes, usually via the blood stream. Blood vessel damage and blockage are often associated with blood infection, as exemplified by mycotic (infective) aneurisms or sub-arachnoid haemorrhage [1]. Indeed, both bacterial and fungal meningitis are associated with vascular events including vasculitis, aneurisms and infarcts [1–5].

The mechanisms of dissemination to the brain in meningitis have been extensively studied *in vitro* and *in vivo*. Experimental studies suggest three potential mechanisms: passage of the pathogen between cells of the blood brain barrier, transcytosis, and passage through the blood brain barrier inside immune cells [6–10]. Here, however, we hypothesise that blood vessel blockage and haemorrhagic dissemination might be an alternative underlying mechanism.

*Cryptococcus neoformans* is an opportunistic fungal pathogen causing life-threatening cryptococcal meningitis in severely immunocompromised patients. *C. neoformans* is a significant pathogen of HIV/AIDs positive individuals with cryptococcal meningitis ultimately responsible for 15% of all AIDS related deaths worldwide [11]. *C. neoformans* has previously been suggested to disseminate from the blood stream into the brain through different routes, including transcytosis, and by using phagocytes as a Trojan horse [6–10]. Consistent with our hypothesis, however, a small number of clinical studies have suggested that blood vessel damage and bursting may also facilitate cryptococcal dissemination. These case reports indicate that cortical infarcts are secondary to cryptococcal meningitis, and suggest a mechanism whereby resulting inflammation may cause damage to blood vessels [12–14]. Notably, retrospective studies of human cryptococcal infection, reported instances of vascular events resulting in infarcts in 30% of cases, predominantly in severe cases of cryptococcal meningitis [15].

There are two major challenges in understanding dissemination during infection. Firstly, the requirement for serial live imaging of a whole animal over hours or days. Secondly, the large variation in microbial pathogenesis and virulence including, but not limited to, hyphal invasion [3], haemolytic toxin production [16] and thrombosis [4]. Long term *in vivo* analysis of infection is not possible in mammalian models; by contrast, the ease of imaging infection in live zebrafish, enables visualisation of infection dynamics over many days [17,18]. Using our zebrafish model of cryptococcosis, we have observed cryptococcal cells becoming trapped and subsequently proliferating within the vasculature. Analysis of the dynamics of infection, via mixed infection of two fluorescent strains of *C. neoformans*, demonstrated that cryptococcal masses within small blood vessels were responsible for overwhelming systemic infection. Localised expansion of *C. neoformans* was observed at sites of dissemination into surrounding tissue. Using a vascular endothelial (VE)-cadherin transgenic reporter line, we identified

physical damage to the endothelial layer in the vasculature at sites of cryptococcal trapping and found that blood vessels respond to their deposition via expansion. Thus, our data demonstrate a previously uncharacterised mechanism of cryptococcal dissemination from the vasculature, through trapping, proliferation, localised blood vessel damage and through a global vasodilation response.

## Results

### Individual cryptococcal cells arrest in blood vessels and form masses

Infection of zebrafish with a low dose of ~25 CFU of *C. neoformans*, directly into the bloodstream, resulted in single cryptococcal cells arrested in the vasculature (Fig 1A). We found that individual cryptococcal cells were almost exclusively trapped in the narrow inter-segmental and brain vessels. It is noteworthy that these vessels are similar in size to mouse brain blood vessels, suggesting a vessel size similar to that of cryptococcal cells may promote trapping (Fig 1B) [19,20]. Studies using intravital imaging in mice have previously noted cryptococcal cell trapping [6], but due to the limitations of long-term imaging in this model the effect of this phenomenon on disease progression could not be established. Exploiting the capacity of zebrafish for long term, non-invasive *in vivo* imaging, we found that the sites of trapped cryptococcal cells proliferated to form cryptococcal masses within blood vessels (Fig 1C, S1 Video). Once cryptococcal masses were established, we found no evidence of their movement along vessels. Furthermore, occlusion by a cryptococcal mass was sufficient to prevent passage of blood cells in blocked inter-segmental vessels (Fig 1D–1E), indicating that blood flow is disrupted. Cryptococcal masses imaged with a cytoplasmic GFP marker did not make direct contact with the vessel wall, due to the presence of the cryptococcal polysaccharide capsule, visualised by antibody staining, which also enveloped large cryptococcal masses (Fig 1F–1H). Thus, we could demonstrate that single cryptococcal cells became trapped in blood vessels and appeared to proliferate to form cryptococcal masses, encased in polysaccharide capsule.

### Clonal expansion of cryptococci in small vessels results in cryptococcal mass formation

We noticed that cryptococcal masses located in vessels appeared early in infection and grew larger overtime (Fig 1C). Examination of infection dynamics over time revealed that cryptococcal masses were often present before overwhelming infection and death (Fig 2A). This suggested to us that the cryptoccocal cells which form masses, being able to survive and proliferate, might be the cryptococcal cells responsible for continued infection and as such represent a population "bottleneck". A population "bottleneck" occurs in infection when the majority of a pathogen inocula is greatly reduced, for example through phagocytosis and degradation, but a minority of the inocula survives. When the surviving minority proliferates, clonality is seen in the downstream infection. Several bacterial pathogens have been demonstrated to establish disease through a population "bottleneck" which leads to clonality of the surviving cells [21,22].

To visualise possible clonal expansion, two different fluorescently labelled cryptococcal strains were compared. The GFP and mCherry labelled *C. neoformans* lines have comparable growth and infection dynamics in the zebrafish model [23]. Initially, we injected a 1:1 ratio of GFP and mCherry-labelled cryptococci and found that single colour infections were very rare. Therefore, we decided to use a higher ratio (5:1) so that we could better quantify the likelihood of clonality arising during the progression of cryptococcal infection. We injected 25 cfu of a

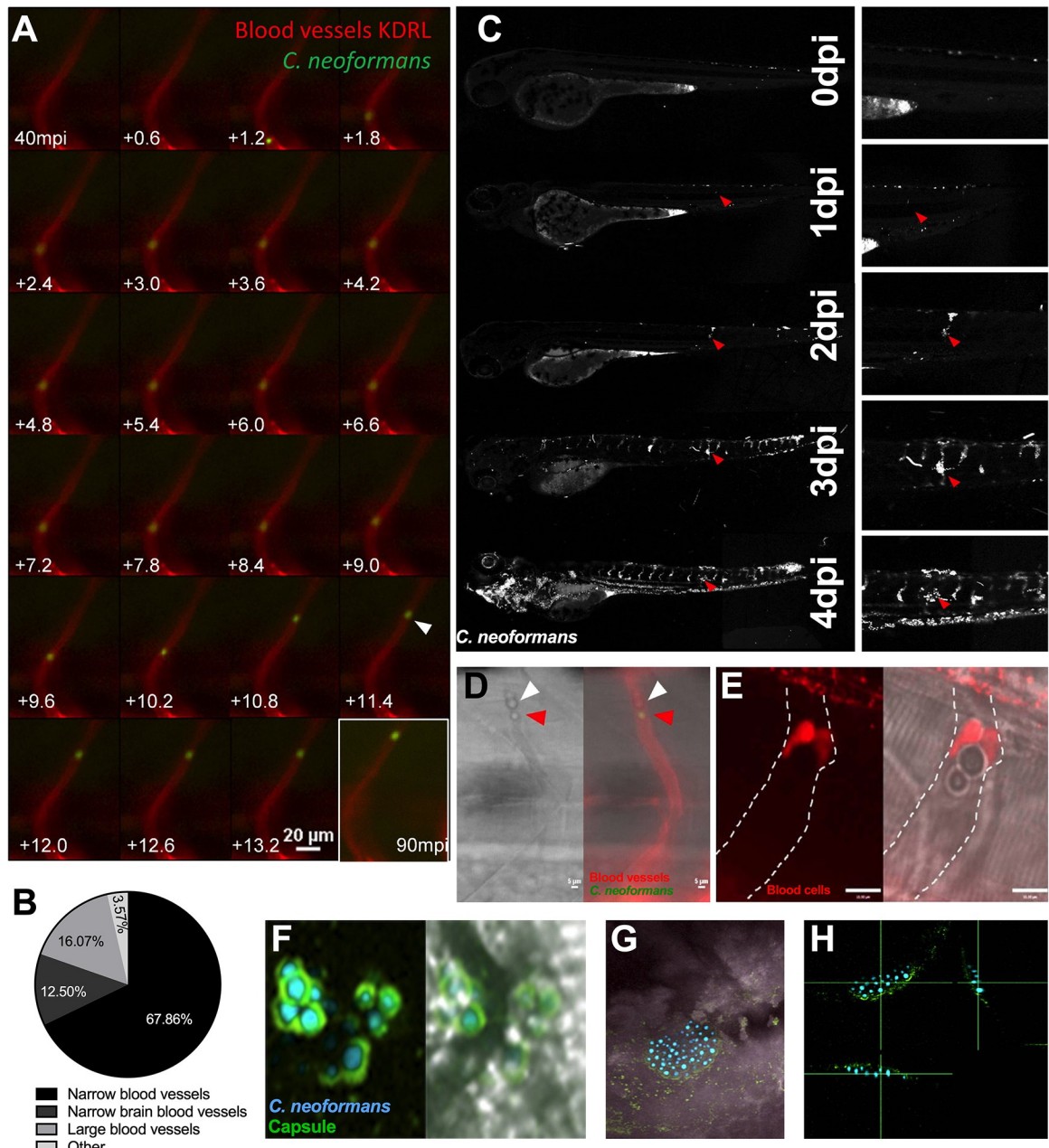

**Fig 1. Cryptococcal mass formation by cryptococcal cell trapping in small blood vessels in the zebrafish.** A Infection of KDRL mCherry blood marker transgenic line with 25 cfu GFP *C. neoformans*, imaged immediately after infection. A single cryptococcal cell becomes trapped in the vasculature (white arrow), at 40 minutes post infection (mpi) after moving from the bottom of the vessel toward the top (left to right, time points +0.6 seconds). Last image shows cryptococcal cell in the same location at the end of the time-lapse at 90mpi B Infection of 2 dpf AB larvae with 25 cfu of a 5:1 ratio of GFP:mCherry KN99 *C. neoformans*. Larvae were imaged until 8 dpf, or death (n = 3, in each repeat 7, 10 and 12 larvae were used) Proportion of cryptococcal masss observed in small intersegmental blood vessels, small brain blood vessels, large caudal vein or in other locations e.g. yolk, (n = 3). C Infection of 2 dpf AB larvae with 25 cfu of a 5:1 ratio of GFP:mCherry KN99 *C. neoformans*. Larvae were imaged until 8 dpf, or death (n = 3, in each repeat 7, 10 and 12 larvae were used). In this case an mCherry majority overwhelming infection was reached. Infection progression from 0 dpi (day of infection imaged 2 hpi), until 4 dpi. Red arrows follows an individual cryptococcal mass formation and ultimate dissemination. D Infection of 2 dpf AB larvae with 25 cfu of a 5:1 ratio of GFP:mCherry KN99 *C. neoformans* showing blood cells (white arrow) trapped behind a cryptococcal mass (red arrow) within an inter-segmental vessel. E Infection of 2 dpf *Tg(gata1:dsRed)* larvae with 1000 cfu GFP of KN99 *C. neoformans* showing blood cells (red) trapped behind a cryptococcal mass within an inter-segmental vessel (white dashed lines). F-H GFP KN99 (cyan), antibody labelled cryptococcal capsule (green). F Cryptococci within blood vessels demonstrating the enlarged capsule blocking the vessel 24 hpi G-H Cryptococcal mass encased in capsule. G Merged florescence and transmitted light z projection H Three-dimensional section of cryptococcal mass showing encasement in polysaccharide capsule.

5:1 ratio of GFP and mCherry-labelled cryptococci and followed the infections for up to 7 days post infection (dpi).

If clonal expansion caused the predominant strain colour at the final end-points (i.e., due to survival and proliferation of a single cryptococcal cell) we would only expect to see single colour end-points. In 51.6% of all infected larvae, a high fungal burden end-point was demonstrated, with cryptococci observed to be either predominantly GFP positive, predominantly mCherry positive, or a mixed outcome of both GFP and mCherry positive (Fig 2B). The remaining 48.4% of infected larvae were able to clear infection (45.2%) or were excluded due to injury in repetitive mounting/imaging (6.5%). Interestingly, a mixed final outcome group was not a rare occurrence (Fig 2C). The high proportion of overwhelming infections caused by mixed GFP and mCherry cryptococci demonstrated that a single cryptococcal cell was highly unlikely to give rise to the final infection population.

The predominantly GFP positive outcome group was observed most often, but only for 56.25% of all high fungal burden endpoints. This was far lower than would be expected, given the initial 5:1 ratio of differently labelled cells injected. While a 5:1 ratio of GFP:mCherry was injected into each larva, the actual number and ratio of cryptococcal cells varied between individual fish (Figs 2D and S1). When single colour and mixed outcomes where compared, there was no significant difference in the injected ratio (Fig 2E), demonstrating the injected ratio does not determine the predominant end-point strain colour. Furthermore, correlative analysis demonstrated that there was no relationship between the initial ratio and final outcome ratio (Fig 2F). Therefore, it appeared that, while a population "bottleneck" caused by a single cryptococcal cell was not responsible for progression of uncontrolled cryptococcal infection, there may be another similar mechanism responsible for promoting survival of cryptococcal strains that contribute to the final population.

Since multiple cryptoccocal masses were present before overwhelming infection (Fig 2A), we hypothesized that selection of cryptococcal strains was determined by the clonal expansion of several cryptococcal masses. To test this hypothesis, we analysed the colours of cryptococcal masses following infection with a 1:1 ratio of GFP and mCherry-labelled cryptococci and found that masses were of a single colour in 14/15 infections at 3 dpi. This suggests that selection of the cryptococcal population within the fish occurs at the cryptococcal mass stage of infection, before the final infection outcome.

We next examined the effect of cryptococcal mass clonal expansion in infection progression and end-points. Cryptococcal masses were observed preceding disseminated infection in every case examined; this process took an average of 2 days (Fig 3A). The maximum number of cryptococcal masses observed within individual larvae was correlated with the rate of infection progression (Fig 3B). This suggests that formation of cryptococcal masses is indicative of infection progression.

We had found that individual cryptococcal cells became trapped in the narrow inter-segmental vessels and brain vessels, which are of similar size to those vessels in the mouse brain where trapping occurs (Fig 1A), and that cryptococcal cells proliferate at these sites (S1 Video, Fig 1C). We quantified the distribution of cryptococcal masses and found that most (80.3%) were located in these smaller brain and inter-segmental blood vessels (Fig 1B). As cryptococcal mass formation at the start of infection was observed in the smaller blood vessels and likely caused by clonal expansion from a single cryptococcal cell (at each site), we determined whether clonal expansion was favoured in smaller blood vessels later in infection. We compared the ratio of GFP:mCherry between the trunk blood vessels and the caudal vein and found that in mixed infections there were single colour masses in the trunk vessels but dual colours in the larger caudal vein (Figs 2A and 3C), suggesting cryptococcal expansion occurs

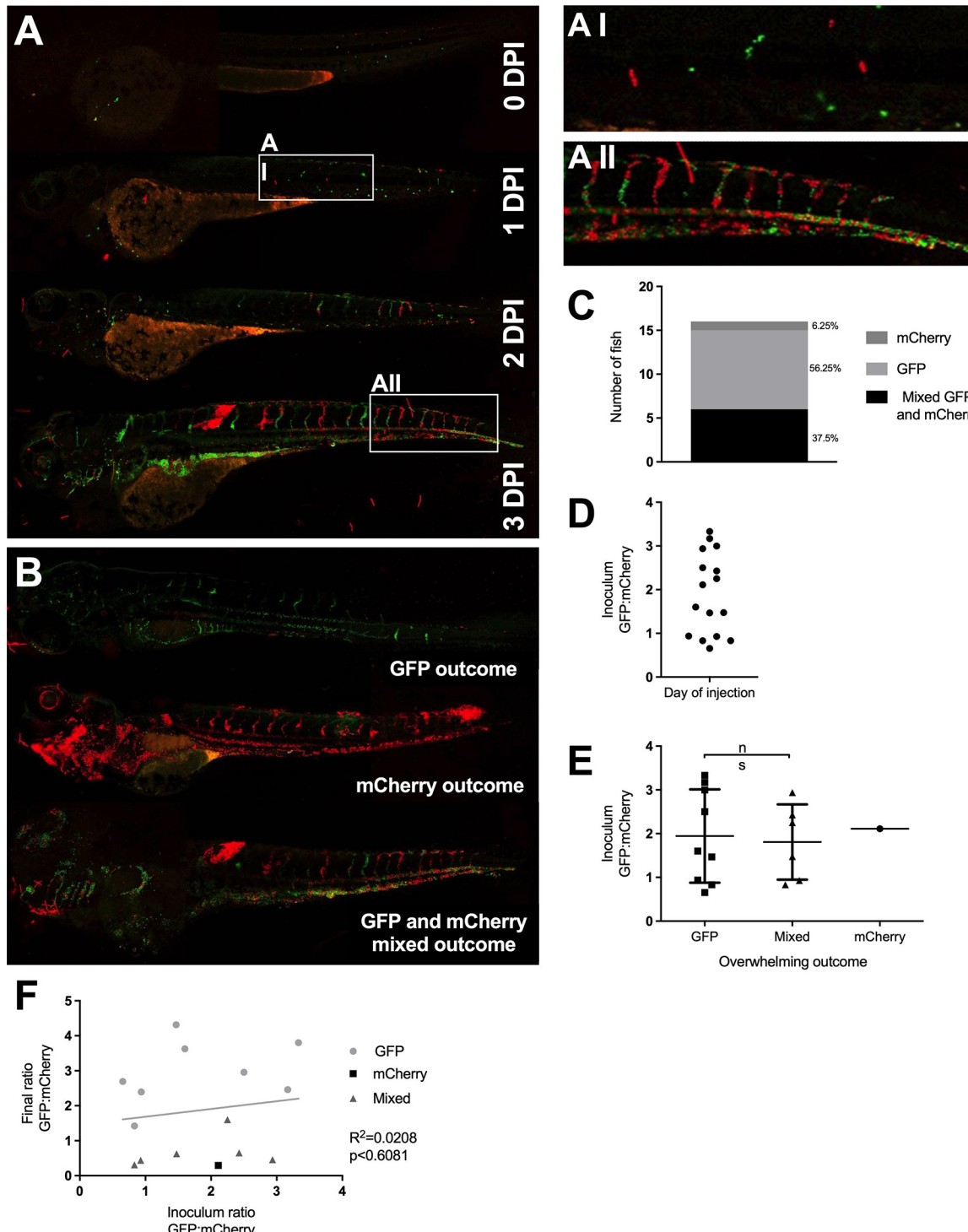

**Fig 2. Inoculum does not predict infection outcome.** Infection of 2 dpf AB larvae with 25 cfu of a 5:1 ratio of GFP:mCherry KN99 *C. neoformans*. Larvae were imaged until 8 dpf, or death (n = 3, in each repeat 7, 10 and 12 larvae were used) A Infection of AB wild-type larvae with 5:1 ratio of GFP:mCherry KN99 *C. neoformans*, at 0 dpi, 1 dpi, 2 dpi and 3 dpi A I Formation of cryptococcal masses at 1 dpi A II Final infection outcome B Infection of 2 dpf AB larvae with 25 cfu of a 5:1 ratio of GFP:mCherry KN99 *C. neoformans*. Larvae were imaged until 8 dpf, or death (n = 3, in each repeat 7, 10 and 12 larvae were used). A GFP majority infection outcome, mCherry infection outcome or a Mixed GFP and mCherry infection outcome (n = 3, 16 larvae) C Proportion of each overwhelming infection outcome observed, GFP, mCherry or mixed D Range of GFP:mCherry *C. neoformans* injected into larvae at 2 hpi E Actual injected GFP:mCherry ratios for each overwhelming outcome (n = 3, +/- SEM, Man-Whitney t-test ns = not significant) F Inoculum ratio of GFP:mCherry, against final GFP:mCherry ratio at overwhelming infection stage (Linear regression $R^2$ = 0.0208, p<0.6081, n = 3, 16 larvae).

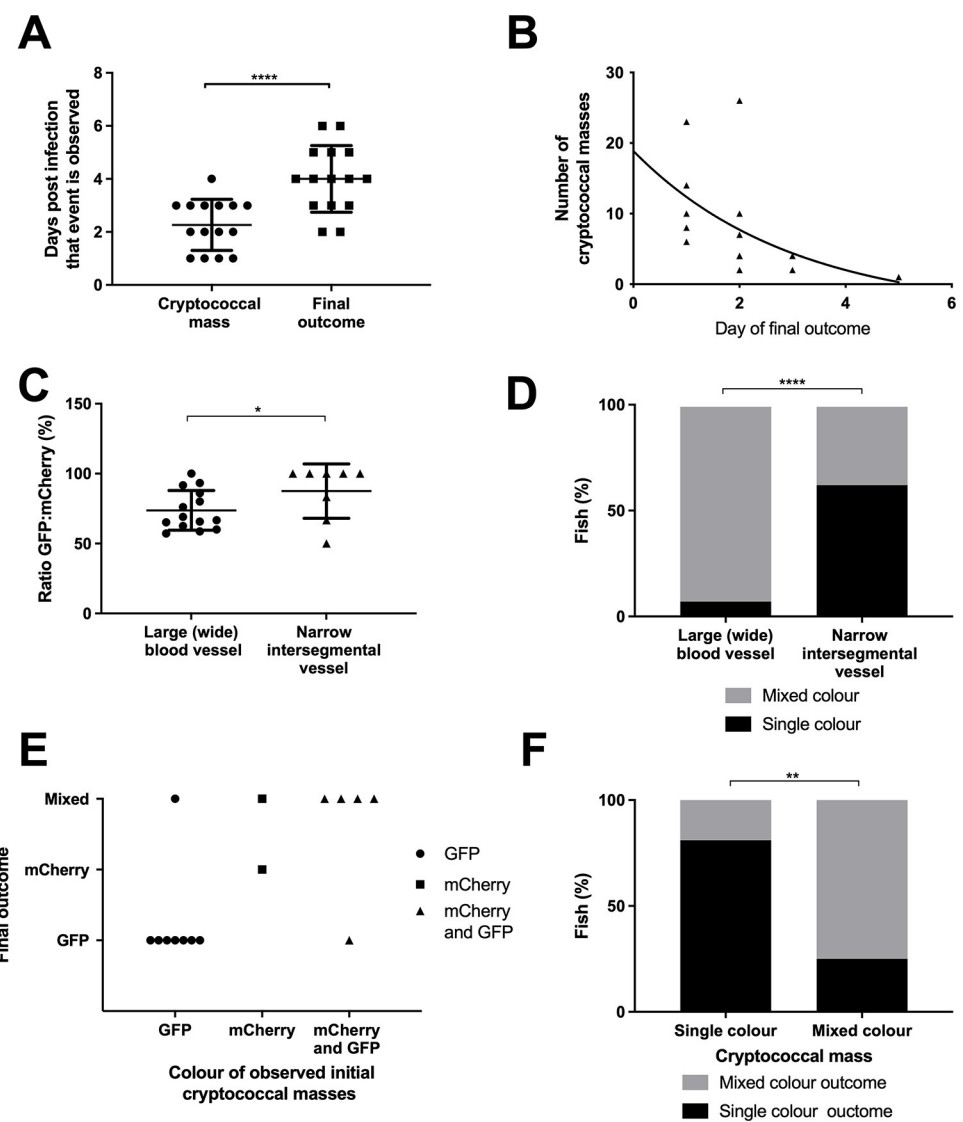

**Fig 3. Cryptococcal mass formation leads to uncontrolled infection.** Infection of 2 dpf AB larvae with 25 cfu of a 5:1 ratio of GFP:mCherry KN99 *C. neoformans*. Larvae were imaged until 8 dpf, or death (n = 3, in each repeat 7, 10 and 12 larvae were used) A Time cryptococcal mass first observed and time of final outcome observed (n = 3, +/- SEM, Wilcoxon matched pairs test, ****p<0.0001) B The maximun number of cryptococcal masss observed within individual larvae and how many days after observation final overwhelming infection was reached (n = 3, non-linear regression, one-phase decay) C The ratio of GFP:mCherry *C.neoformans* in the large caudal vein in comparison to the fifth intersegmental blood vessel, at uncontrolled infection time point (n = 3, *p<0.05, +/-SEM, paired t-test). D Single or mixed *C.neoformans* strains in the large caudal vein in comparison to the fifth inter-segmental blood vessel, at uncontrolled infection time point (n = 3, ****p<0.0001, Fischer's exact test). E Comparison of the colour (either GFP, mCherry or mixed) of *C. neoformans* in cryptococcal masses found in inter-segmental vessels, in relation to the final outcome majority *C. neoformans* colour F Comparison of the colour of cryptococcal masss, either single colour or mixed, with the colour of final outcome (n = 3, **p<0.01, Fischer's exact test).

at sites of trapping in narrow blood vessels. Comparison of strain colour in these blood vessels showed strains in narrow blood vessels were significantly more often a single colour (Fig 3D).

Finally, we examined whether cryptococcal mass strain colour predicted the predominant strain colour at the end-point of infection. We compared the colours of individual cryptococcal masses with majority colour of high fungal burdens within individual fish.

If individual cryptococcal masses found clonal colonies that spread through the larvae to influence the predominant end-point strain, we would expect a correlation; for example presence of a GFP cryptococcal mass would be likely to precede a GFP predominant strain final outcome. Indeed, a clear relationship was demonstrated between the colour of each cryptococcal mass and the disseminated infection; a single (GFP or mCherry) cryptococcal mass colour was significantly more likely to result in a single colour final outcome, with a corresponding finding for mixed cryptococcal masses (Fig 3E and 3F p<0.01). Together these observations suggested cryptococcal cells became trapped in small blood vessels followed by localised clonal expansion, which influence the final cryptococcal population.

## Cryptococcal masses cause local and peripheral vasodilation

The finding that cryptococcal masses blocked blood vessels prompted us to measure blood vessel width at sites with or without cryptococcal masses. We found that blood vessels that contained cryptococcal cells were significantly wider than those devoid of cryptococcal cells in the same infections (Fig 4A and 4B). A higher infection dose was used to increase the number of cryptococcal masses that formed for analysis. There was a significant difference very early, at 2 hours post infection (hpi), and a much larger difference at 3 dpi (Fig 4A and 4B), suggesting an immediate effect (i.e. due to the elasticity of the vessel wall) and a slower physical widening of the vessel caused by cryptococcal growth. We explored the first surmise of a fast response of the blood vessel by live imaging small brain vessels and observed, through measurement of vessel width, that vessels locally dilated shortly after blockages formed (Fig 4C). The increase in vessel width was proportional to the size of the cryptococcal mass inside the vessel at both 2 hpi and 3 dpi (Fig 4D and 4E) suggesting a slow increase in vessel width occurs over time due to growth of the cryptococcal mass pushing against the vessel wall. Injection of inert beads of a size corresponding to an average cryptococcal cell (4.5 μm) did not lead to formation of large masses, although there was a small but significant increase in vessel size at locations where beads did become trapped in the vasculature by 3 dpi (Fig 4F). Beads were observed to be stuck in the inter-segmental blood vessels much less frequently than live cryptococcal cells, with 13.6% of blood vessels containing beads compared to 89.0% containing cryptococcal cells. In addition to examining inter-segmental vessels, we specifically imaged the small vessels of the brain and found that blood vessels containing cryptococcal cells were larger relative to blood vessels in the same location in non-infected animals (Fig 4G and 4H). Thus, it appeared that blockage by cryptococcal cells and masses increased vessel diameter due to immediate responses and slower changes (due to cryptococcal growth) in blood vessels. The immediate response is likely to reduce the total peripheral resistance, which is increased in cryptococcal infection and appears similar to the role of increased peripheral resistance and vessel tension in higher frequencies of aneurysm [24]

## Cryptococcal cell size affects the frequency of blood vessel occlusion

In order to investigate whether cryptococcal cell size or rigidity may affect the frequency of trapping and the extent of blood vessel vasodilation, we used mutant cryptococci with altered physical properties. Recently, the biophysical properties of several ceramide pathway mutants have been described [25] in which the accumulation of saturated GluCer (*Δsld8*) was suggested to increase the rigidity of cryptococcal membranes and therefore reduce their ability to traverse smaller blood vessels. In contrast, mutants in *Δgcs1* and *Δsmt1* have reduced amounts of the more rigid ceramide lipids or differences in lipid packing respectively. Therefore, we predicted that the *Δsld8* mutant would produce an increased number of blocked vessels whereas the *Δgcs1* and *Δsmt1* would produce reduced numbers of blockages. However, we found no

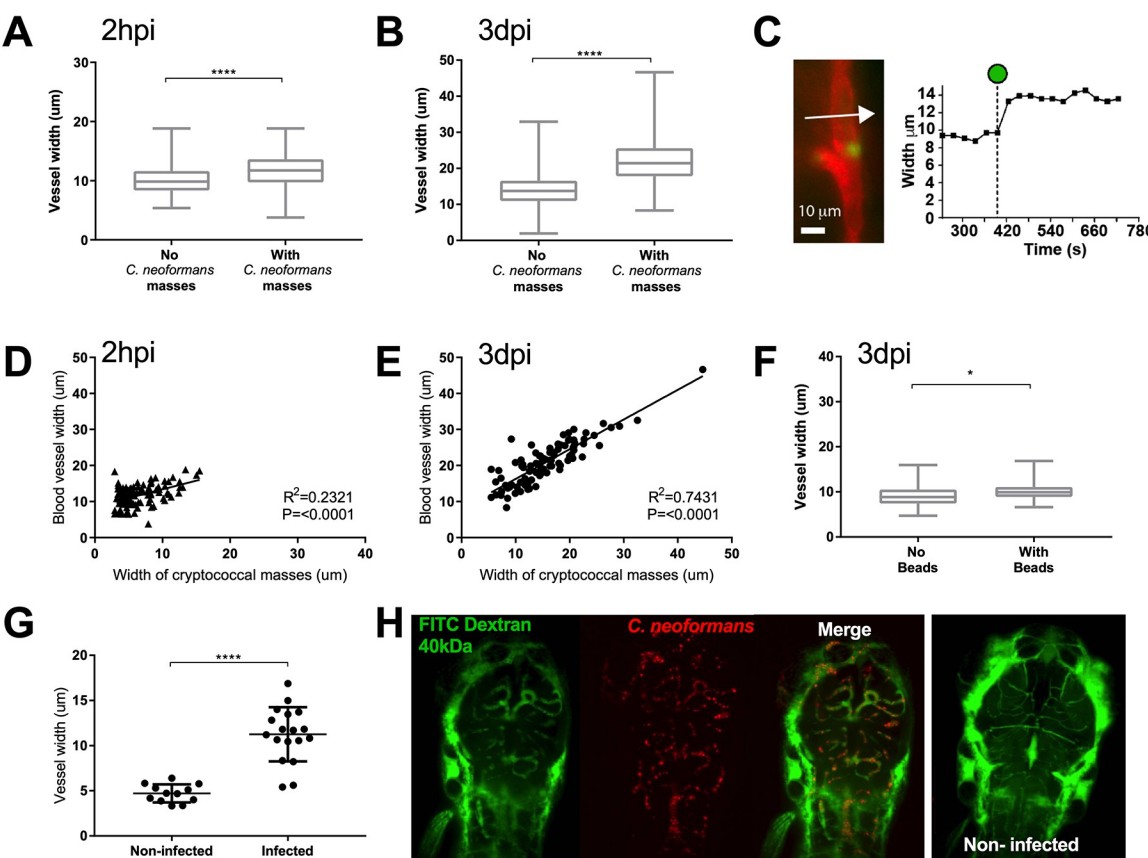

**Fig 4. Localised clonal expansion proportionally increases vasculature size.** A-E: Infection of KDRL mCherry blood marker transgenic line with 1000 cfu GFP *C. neoformans* or inert beads A Inter-segmental vessel width with and without cryptococcal masses at 2 hpi (n = 3, +/- SEM, ****p<0.0001, unpaired t-test) B Inter-segmental vessel with and without cryptococcal masses at 3 dpi (n = 3, +/- SEM, ****p<0.0001, unpaired t-test) C Left panel—Image from a time-lapse movie of KDRL mCherry zebrafish larvae showing a blood vessel (red) in the zebrafish brain and a *C. neoformans* cell (green). Right panel—graph showing the change in diameter of the blood vessel measured at the point indicated by the white arrow in Ci, at each frame in the time-lapse. The dotted line on the x axis indicates the timepoint where the *Cryptococcus* cell becomes stuck at the point of measurement (white arrow). D Relationship between *C. neoformans* mass and vessel width at 2 hpi (n = 3, linear regression) E Relationship between *C. neoformans* mass and vessel width at 3 dpi (n = 3, linear regression) F Vessel width with and without beads present at 3 dpi (n = 3, +/- SEM, *p<0.05, unpaired t-test). G-H: Inoculation of mCherry *C. neoformans* with 40kDa FITC Dextran to mark blood vessels G Comparison of infected brain vessels width to non-infected corresponding brain vessels (three infected fish analysed, +/- SEM, ****p<0.0001, paired t-test) H Example image of infected and non-infected brain vessels.

differences in the number of blocked vessels or vessel width in either *Δgcs1*, *Δsmt1* or *Δsld8* compared to their reconstituted strains (S2–S4 Figs).

Next, we asked whether fungal cell sized altered blockage and dilation of blood vessels. Deletion of *Δplb1* has previously been shown to exhibit increased cell size during infection of macrophages *in vitro* and in a mouse model of cryptococcosis [26,27]. Although a *Δplb1* mutant was less able to disseminate into the brain from the blood in rabbits [28], cryptococcal strains expressing reduced *plb1* can disseminate from the blood into the brain in mice [29]. Furthermore, *plb1* is required for escape from lungs, not for dissemination from the blood into the brain [30]. We have recently demonstrated that several phenotypes associated with *plb1* deletion were due to differences in fungal eicosanoid production, differences also present in a second cryptococcal mutant strain *Δlac1* [31]. A role for *lac1* in promoting cryptococcal

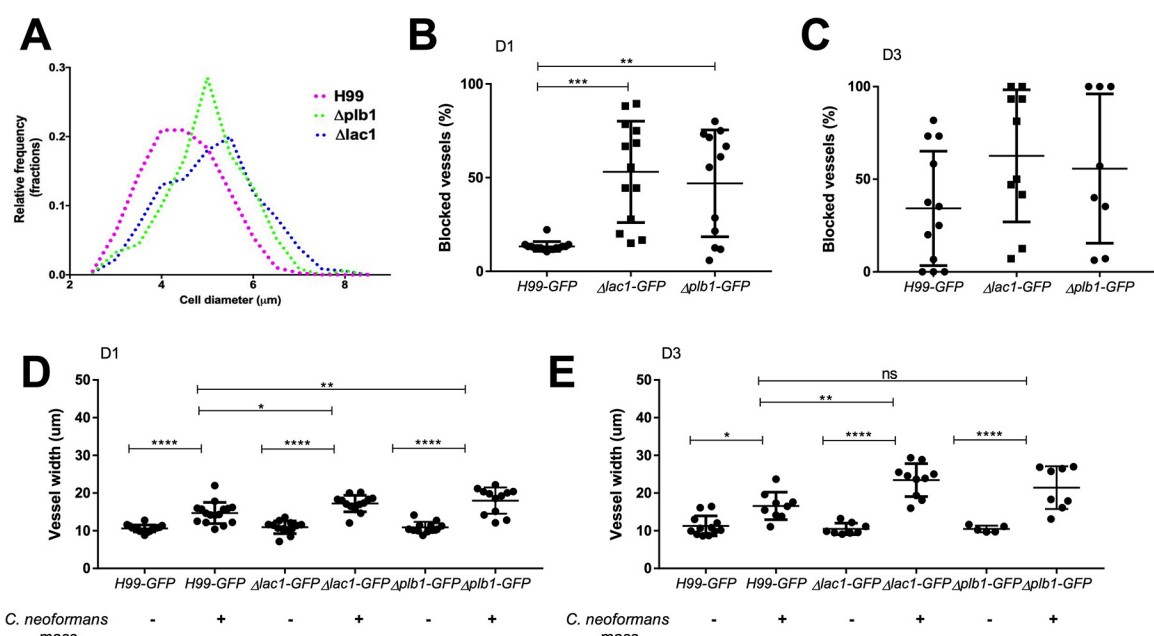

**Fig 5. Cryptococcal cell size influences the frequency of trapping within blood vessels.** A-E: Infection of KDRL mCherry blood marker transgenic line with 1000 cfu *Δplb1-H99*, *Δlac1-H99* or parental *H99-GFP C. neoformans* A Size of cryptococcal cells injected into zebrafish larvae on the day of infection (>300 cryptococcal cells measured per strain) B Blocked vessels (% of all inter-segmental vessels) at 1 dpi (n = 2, +/- SD, **p<0.01, Kruskal-Wallis test) C Blocked vessels (% of all inter-segmental vessels) at 3 dpi (n = 2, +/- SD, Kruskal-Wallis test) D Vessel width with or without *C. neoformans* at 1 dpi (n = 2, +/- SD, ns = not significant, **p<0.01, ****p<0.0001, Kruskal-Wallis test) E Vessel width with or without *C. neoformans* at 3 dpi (n = 2, +/- SD, ns = not significant, *p<0.05, ****p<0.0001, Kruskal-Wallis test).

dissemination has been described, suggesting that *lac1* is important for lung dissemination, but like *plb1*, *lac1* is not required for dissemination into the brain [32,33]. We first wanted to ascertain whether enlarged cryptococcal size also occurs in the zebrafish infection model; we measured the size of *Δplb1* and *Δlac1* cryptococcal cells at 1 dpi, and found that there was a significant increase in cell diameter compared to wild-type, with a 100% increase in the number of cryptococci with a diameter >5 μm (Fig 5A). As human, rodent and zebrafish capillaries are close to 5 μm at their smallest, we hypothesized that the increased fungal cell diameter of the *Δplb1* and *Δlac1* mutant cells would increase the number of vessels that would be blocked by cryptococci.

We counted the number of blocked vessels in infections with wild-type, *Δplb1* and *Δlac1* mutant cryptococci and found there was a large increase in the proportion of blocked vessels at 1 dpi; in some cases more than 80% of inter-segmental vessels were blocked by the *Δplb1* and *Δlac1* mutant cells (Fig 5B). The difference in the proportion of blocked vessels was no longer significant by 3 dpi (Fig 5C). We also measured the width of vessels and found a small but significant increase for *Δlac1* and *Δplb1* at 1 dpi in comparision to controls, with *Δlac1* significantly increased vessel width at 3 dpi (Fig 5D and 5E). Therefore, increased cryptococcal cell diameter were associated with an increase in the frequency of vessel blockages. This is in addition to having a small but significant influence increasing vessel diameter at sites of masses at early timepoints. These effects are likely both due to the larger cell size of *Δlac1* and *Δplb1* cells increasing the possibility of trapping in larger vessels, as well as directly having a larger impact on vessel width once trapped.

## Cryptococcal infection increases blood vessel tension resulting in hemorrhagic dissemination

Following long-term time-lapse imaging of cryptococcal masses, we observed that enlargement of the cryptococcal mass over time eventually led to invasion of the surrounding tissue at the site of infection (Fig 1C). The mechanism by which cryptococci disseminate from blood vessels is unknown, but has been suggested to be via transcytosis or within immune cells *in vitro* [6–10]. However, from our observations and from clinical reports linking vasculature damage with cryptococcal meningitis and cortical infarcts [12–14], we hypothesised that cryptococcal masses were blocking vessels, increasing the force on the blood vessel walls, leading to vessel damage, rupture and eventual dissemination of cryptococci. To test our hypothesis, we first established the association between tissue invasion and sites of clonal expansion within the vasculature. We found that in all cases tissue invasion occurred at sites of clonal expansion within the vasculature (19/19 tissue invasion events observed from 29 infected zebrafish; Fig 6A). Furthermore, *C. neoformans* that had invaded the surrounding tissue were invariably the same colour (GFP or mCherry) as the closest vasculature cryptococcal mass (Fisher's exact test p<0.001, n = 3, Fig 6B). To determine whether the vasculature was physically damaged sufficiently for cryptococcal cells to escape into the surrounding tissue, we examined blood vessels at high resolution at the sites of tissue invasion. We observed vessel widening and damage, bursting and cryptococcal dissemination at locations of cryptococcal masses (Fig 6C-6E), including dissemination where the vasculature remained intact (Fig 6C), but never in non-infected vessels (Fig 6E).

To examine possible damage of blood vessels at sites of cryptococcal masses, we injected dextran into the bloodstream 3 days after they were infected with *C. neoformans* at 2 dpf, to visualize potential leakage at sites of vessel blockage. Dextran was not present to the same extent in blocked inter-segmental vessels in comparison to open vessels (Fig 6F), supporting our previous findings that cryptococcal masses appeared to block blood flow (Fig 1). Small amounts of dextran were rarely observed outside blood vessels, being observed only at sites of major cryptococcal dissemination (Fig 6G). Conceivably, this is likely due to the lack of blood flow enabling dextran to enter blocked vessels.

To investigate vessel damage, with visualisation of fluorescent vessel markers, we used a double transgenic line to mark vascular endothelial cells expressing junctional protein VE cadherin (*Tg*(*10xUAS*:*Teal*)*uq13bh*) [34] and the endothelial *TgBAC*(*ve-cad*:*GALFF*) driver [24]. Using these transgenic fish, we also found that cryptococcal cells were located outside the blood vessel when vessels were either intact (fluorescence observed along the vessel walls at the site of a cryptococcal mass, suggesting the blood vessel endothelial layer in intact) (Fig 6H) or disrupted (fluorescence was not observed for the full vessel walls at the site of a cryptococcal mass, suggesting damage in the blood vessel endothelial layer) (Fig 6I), in comparison to non-infected vessels (Fig 6J). However, depleting macrophages from larvae with clodronate treatment led to increased dissemination of cryptococcal cells into somite tissue, indicating that macrophages are not required for dissemination events (Fig 6K). Furthermore, co-localisation of phagocytes at cryptococcal masses occurred in only 25% of small (up to four cells) masses, and 32% of larger masses observed (Figs 6L, 6M and S5). In many cases phagocytes interacted with just a small section of the cryptococcal mass (S5C Fig).

We measured VE-cadherin intra-molecular tension at cell-cell junctions between vascular endothelial cells using the FRET reporter, the zebrafish transgenic line *TgBAC*(*ve-cad*:*ve-cadTS*)*uq11bh* (hereafter VE-cadherin-TS) [24]. As previously described [24], the ratio of the FRET excitation of YFP (FRET signal) intensity to the intensity of independent direct laser excitation of YFP is a measure of the distance between VE-cadherin adhesion proteins and

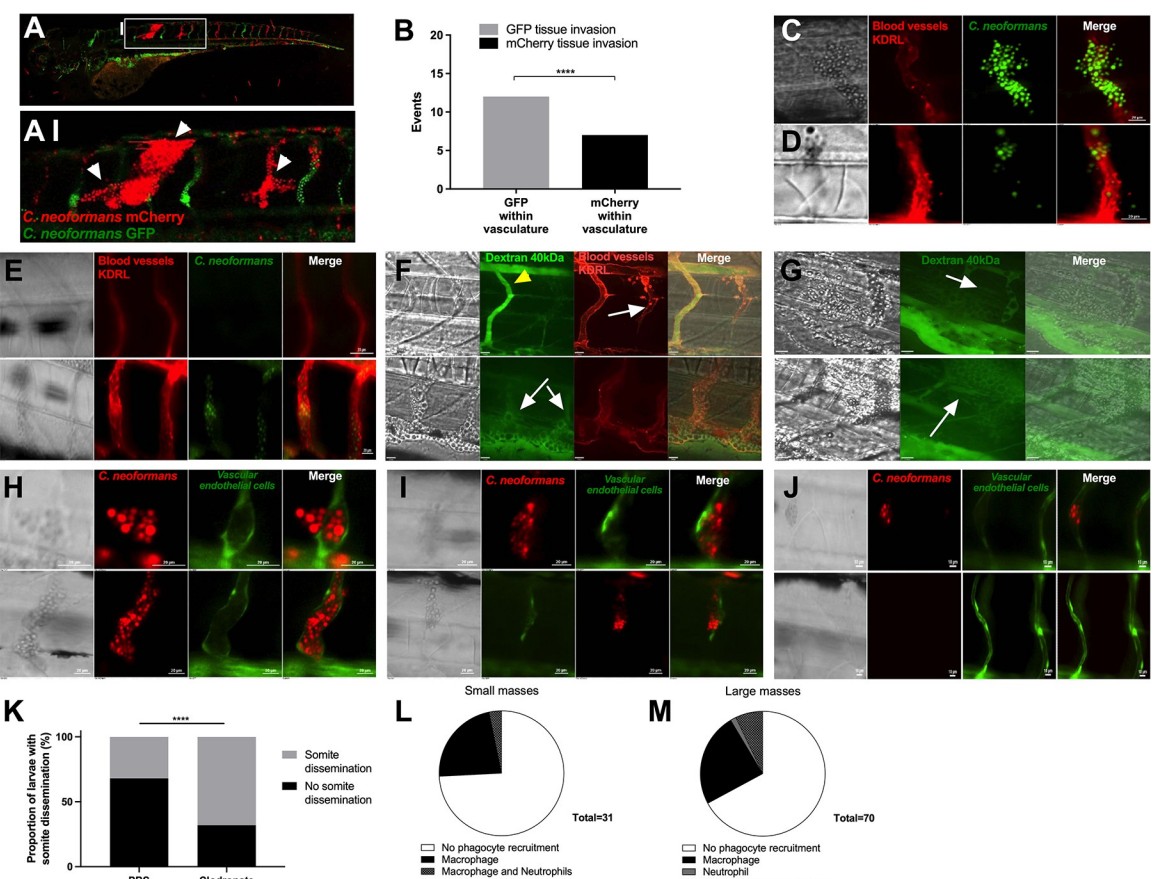

**Fig 6. Dissemination events through vasculature damage.** A-B Infection of 2 dpf AB larvae with 25 cfu of a 5:1 ratio of GFP:mCherry KN99 *C. neoformans.* Larvae were serially imaged until 8 dpf, or death A-AI white arrow heads indicate examples of dissemination of *C. neoformans* (mCherry) into the somite surrounding an existing mCherry cryptococcal mass B Comparison of colour of *C. neoformans* in the vasculature (GFP or mCherry), and the corresponding colour of dissemination events at the same location (19 events in total) C, D and E Infection of KDRL mCherry blood marker transgenic line at 2 dpf with 1000 cfu GFP *C. neoformans* C Dissemination from an intact blood vessel, with *C. neoformans* in the surrounding tissue suggested to be transcytosis D Damaged blood vessels with *C. neoformans* in surrounding tissue E Intact blood vessels (KDRL marker) with or without *C. neoformans* F-G Infection of KDRL mCherry blood marker transgenic line at 2 dpf with 1000 cfu *C. neoformans* and then injection of 40kDa FITC Dextran (green) at 5 dpf immediately before imaging F yellow arrowhead indicates dextran within unblocked inter-segmental vessel, white arrows indicate cryptococcal masses within inter-segmental vessels which do not have dextran within. G white arrows indicate sites of dextran leakage into surrounding somite next to cryptococcal dissemination events. H-J Infection of vascular-endothelium cadherin endothelial junction (blood vessel marker) transgenic line with 1000 cfu mCherry *C. neoformans* H Intact blood vessel endothelial layer, with *C. neoformans* in the surrounding tissue I Damage in the blood vessel endothelial layer J Intact blood vessels with or without *C. neoformans* K Proportion of larvae developing cryptococcal somite growths by 3 dpi, infected with 500 cfu H99-GFP *C. neoformans* with clodronate liposome or PBS-control treatment (n = 3, groups of 92 and 145 larvae). L-M Infection of *Tg(mpeg1:mCherry.CAAX)sh378* stably crossed to *Tg(mpx:eGFP)i114* larvae at 2 dpf with 1000 cfu KN99 *C. neoformans* imaged at 3 dpi L The number of small cryptococcal masses (up to 4 cryptococcal cells) with macrophage, neutrophil, both or no phagocyte recruitment (n = 3, 26 larvae) M The number of larger cryptococcal masses (over 4 cryptococcal cells) with macrophage, neutrophil, both or no phagocyte recruitment (n = 3, 26 larvae). In this figure images of vessels show a single plane of a 50 μm stack.

therefore an indirect measure of the vessel tension (i.e. increased tension results in a decrease in FRET signal due to an increase in distance between VE-cadherin molecules). Measuring the FRET/YFP ratio we found a clear decrease in FRET signal between infected and uninfected zebrafish, indicating an increased tension in the vessels during cryptococcal infection (Fig 7A–7C). In addition, we found that the FRET signal was decreased in both vessels with cryptococcal masses and those without, supporting our previous data demonstrating a global

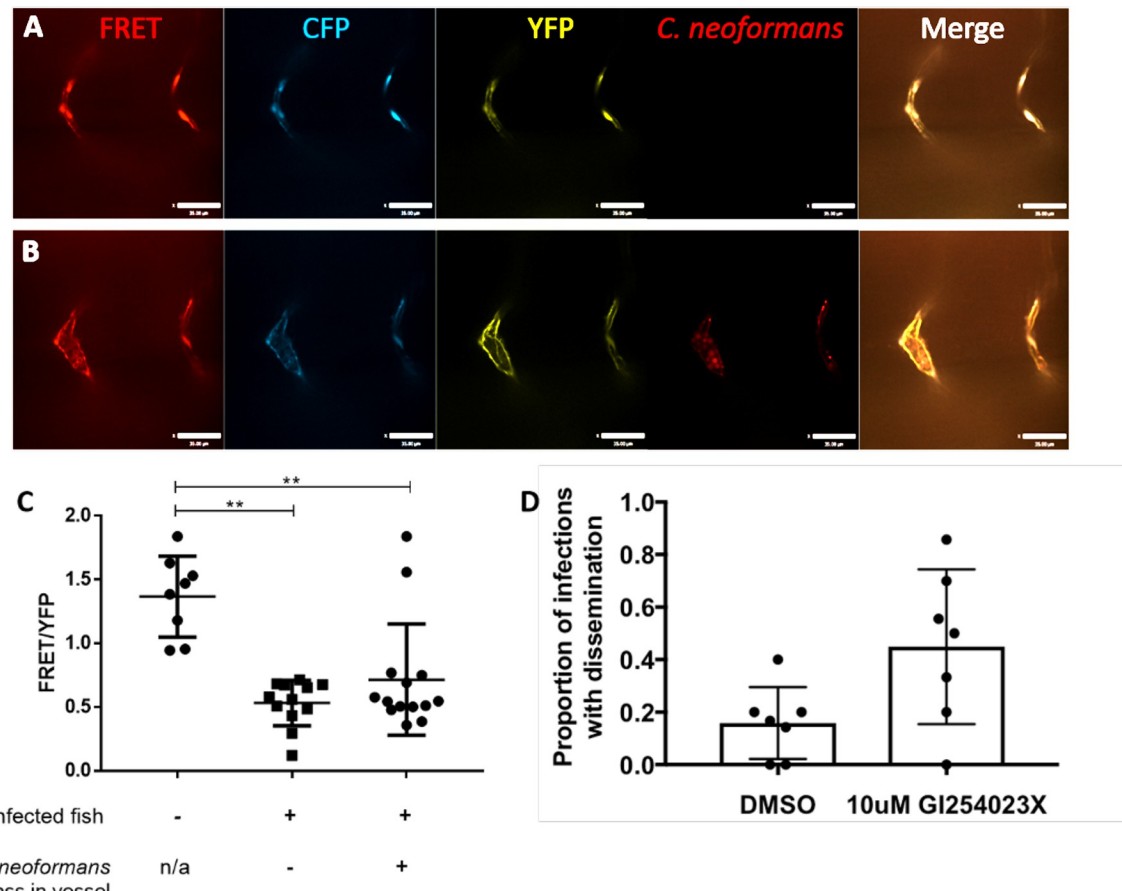

**Fig 7. Cryptococcal infection leads to increased tension across VE-cadherin.** A-C Infection of FRET tension reporter (VE-cadherin-TS) transgenic zebrafish line with 1000 cfu mCherry *C. neoformans* A-B FRET = YFP signal via FRET with excitation of CFP, CFP = CFP signal with direct excitation of CFP, YFP = YFP signal with direct excitation of YFP. A images of non-infected control vessels B Image showing infected fish, vessel containing a mass (left) and a vessel without a mass (right) C FRET analysis of infected fish with or without masses and non-infected controls larvae (n = 2, 4–7 larvae per repeat, +/- SD, **p<0.01, Kruskal-Wallis test, where vessel fluorescence was measured at each side of vessel) D Proportion of infected fish with disseminated infection. 7 repeats, 10 zebrafish larvae per repeat per group. P = 0.036 unpaired t-test.

vasodilation in response to increased peripheral resistance (Fig 7C). These data suggested that there was an increase in vessel tension associated with cryptococcal growth in vessels. Recently it has been shown that aneurysms have a higher chance of rupture under high vessel tension and peripheral resistance [24]. Therefore, to test if the increased peripheral resistance was causing the haemorrhagic dissemination we had observed, we sought to increase peripheral resistance and vessel stiffness simultaneously by inhibiting the elasticity of blocked vessels. VE-cadherin is regulated extracellularly by the protease ADAM10 and one effect of ADAM10 inhibition is increased VE-cadherin junctions and blood vessel stiffness [35]. Therefore, we used an ADAM10 inhibitor (GI254023X, a competive inhibitor with 100-fold higher specificity for ADAM10 versus ADAM17 [36]) during infection to increase peripheral resistance and vessel stiffness simultaneously. Consistent with our prediction, we found that inhibition of ADAM10 was sufficient to cause a large increase in the number of haemorrhagic dissemination events (Fig 7D). This indicates a dual route of cryptococcal dissemination caused by cryptococcal masses located in vessels, through a general increased peripheral resistance and vessel tension; as well as through localised physical damage to blood vessels.

## Discussion

Here we have demonstrated how *C. neoformans* can cause haemorrhagic dissemination from blood vessels, suggesting a generalised mechanism for infarct formation during infective meningitis. Consistent with post-mortem reports showing pathogens in the brain located next to capillaries with fungal or bacterial masses present [37], our data demonstrate that, even at very low levels of fungemia, cryptococci can form masses in blood vessels, leading to increased vessel tension and blood vessel haemorrhage. This dissemination mechanism is of importance for understanding progression of human cryptococcal meningitis infections and associated infarct formation previously described in clinical case reports [12–14]. We show both localised vessel and a global vasodilation response during infection. Localised vessel vasodilation and associated damage is caused by pathogen proliferation; however the global response is likely caused by an increase in the peripheral resistance. We demonstrate that proliferation of trapped fungal cells leads to skewing of the fungal population. Importantly, we demonstrate vessel damage occurs at sites of cryptococcal masses, which likely leads to cryptococcal escape to the surrounding tissue. However, we observed dissemination via transcytosis (demonstrated where vessel structure was still intact) at sites of cryptococcal masses, which is suggestive that transcytosis events are also promoted by the presence of large masses of cryptococcal cells. This implies that cryptococcal growth within blood vessels facilitates dissemination events not only through vasculature damage.

Vascular damage following fungal infection may not be limited to *C. neoformans*; clinical reports show invasion of blood vessels by *Aspergillus fumigatus* could lead to hemorrhagic infarct formation [3] and in meningitis caused by *Coccidioides immitis*, infarcts were observed, at locations of thrombosis [4], potentially caused by fungal cell blockage of vessels. Furthermore, haemorrhage has been observed following an aneurysm caused by *Mucor* infection in an immuno-compromised patient with primary mucormycosis [38]. Similarly, mycotic aneurysm, vasculitis and also blood vessel occlusion were observed in zygomycosis infection [39], suggesting a trapping of fungal cells and blood vessel damage may occur in different fungal species infection.

Furthermore, we demonstrate that following cryptococcal infection a global increase in vessel vasodilation and tension across VE-cadherin occurs within the larvae, likely due to increased blood flow in unblocked vessels. We suggest increased blood flow, and therefore likely increased vessel tension, (supported with increased tension in VE-cadherin molecules) progress a positive feedback loop of increased blockages leading to further increased blood flow, vessel tension and ultimately dissemination. Indeed, we demonstrate that increasing vessel stiffness leads to increased dissemination events. This may be similar to the observed increased risk of aneurysm with high vessel tension and peripheral resistance without infection [24], perhaps further enhanced as infection progresses. In bacterial meningitis local damage occurs to vascular endothelial cells, but interestingly there is also an imbalance of hemostatic forces, potentially caused by multiple immune responses to infection, which may have a systemic effect similar to what we have shown here [40]. This suggests that blood flow and vessel tension are important factors in multiple vascular infections and diseases.

The mechanism of haemorrhagic dissemination we have described for *C. neoformans* may be relevant to many infections, with multiple pathogens known to cause infarcts and vasculitis in human infection [41,42]. In addition, infarcts are also observed in meningitis caused by bacterial pathogens, for example *S. enterica* and *T. bacillus* [43,44]. Tuberculosis meningitis can also cause vasculitis leading to infarct formation [2]. In bacterial meningitis, caused by *N. meningitidis*, the level of bacteraemia causes different types of vascular damage. At low bacterial numbers, bacteria are able to colonise brain blood vessels and cause limited vessel damage,

eventually leading to meningitis. In contrast, high bacterial load is associated with increased vascular colonisation and augmented vascular damage [1], indicating that higher blood vessel blockage can cause increased blood vessel damage, perhaps in a similar positive feedback loop as we suggest for cryptococcal meningitis.

Thus, the novel mechanism of cryptococcal dissemination that we have demonstrated may be the physiological cause of infarcts observed in blood infection. Pathogen cell trapping in narrow blood vessels, based on size, leads to localised proliferation. Growth leads to blood vessel vasodilation and damage which can allow cryptococcal cell escape into the surrounding area. In addition, cryptococcal infection induces a global vasodilation response which is associated with increased vessel tension and dissemination events. Our proposed mechanism for blood vessel bursting in cryptococcal infection may exist for other pathogens which cause vascular damage or haemorrhages, and vary depending on individual pathogen traits.

## Methods and methods

### Ethics statement

Animal work was carried out according to guidelines and legislation set out in UK law in the Animals (Scientific Procedures) Act 1986, under Project License PPL 40/3574 or P1A4A7A5E. Ethical approval was granted by the University of Sheffield Local Ethical Review Panel. Animal work completed in Singapore was completed under the Institutional Animal Care and Use Committee (IACUC) guidelines, under the A*STAR Biological Resource Centre (BRC) approved IACUC Protocol # 140977.

### Fish husbandry

Zebrafish strains were maintained according to standard protocols [45]. Animals housed in the Bateson Centre aquaria at the University of Sheffield, adult fish were maintained on a 14:10-hour light/dark cycle at 28˚C in UK Home Office approved facilities. For animals housed in IMCB, Singapore, adult fish were maintained on a 14:10-hour light/dark cycle at 28˚C in the IMCB zebrafish facility. We used the *AB* and *Nacre* strains as the wild-type larvae. The blood vessel marker *Tg(kdrl:mCherry)s916* and red blood cell marker *Tg(gata1:dsRed) [46],* in addition to *Tg*(*10xUAS*:*Teal*)*uq13bh* crossed to endothelial *TgBAC*(*ve-cad*:*GALFF*) [24] for stable expression. We used the FRET tension sensor line, *TgBAC*(*ve-cad*:*ve-cadTS*) *uq11bh* [24]. We also used an mCherry macrophage line, *Tg(mpeg1:mCherry.CAAX)sh378* [17], stably crossed to a GFP neutrophil line, *Tg(mpx:eGFP)i114* [47], to examine both cell types in the same larvae.

### *C. neoformans* culture

The *C. neoformans* variety *grubii* strain KN99, its GFP-expressing derivative KN99:GFP and mCherry-expressing derivative KN99:mCherry were used in this study [23]. We used GFP expressing *Δplb1-H99*, *Δlac1-H99* or parental *H99-GFP* [31] *and Δgsc, Δsmt, Δsld8 and parental strain* [25]. Cultures were grown in 2 ml of yeast extract peptone dextrose (YPD) (all reagents are from Sigma-Aldrich, Poole, UK unless otherwise stated) inoculated from YPD agar plates and grown for 18 hours at 28˚C, rotating horizontally at 20 rpm. Cryptococcal cells were collected from 1ml of the culture, pelleted at 3300 g for 1 minute.

To count cryptococcal cells, the pellet was re-suspended in 1 ml PBS and cells were counted with a haemocytometer. Cryptococcal cells were pelleted again at 3300g and re-suspended in autoclaved 10% Polyvinylpyrrolidinone (PVP), 0.5% Phenol Red in PBS (PVP is a polymer that increases the viscosity of the injection fluid and prevents settling of microbes in the

injection needle), ready for micro-injection. The volume cryptococcal cells were re-suspended in was calculated to give the required inoculum concentration.

## Zebrafish microinjection

Zebrafish larvae were injected at 2 days post fertilisation (dpf) and monitored until a maximum of 10 dpf. Larvae were anesthetised by immersion in 0.168 mg/mL tricaine in E3 and transferred onto 3% methyl cellulose in E3 for injection. 1nl of the suspension of cryptococcal cells, where 1nl contained 25 cfu, 200 cfu or 1000 cfu, was injected into the yolk sac circulation valley. For micro-injection of GFP fluorescent beads (Fluoresbrite YG Carboxylate Microspheres 4.50μm). The bead stock solution was pelleted at 78g for 3 minutes, and re-suspended in PVP in phenol red as above for the required concentration. Micro-injection of 40kDa FITC-dextran at 3 dpf in a 50:50 dilution in PVP in phenol red, injected 1 nl into the duct of Cuvier. For micro-injection of 40kDa FITC-dextran at 5 dpf, dextran was resuspended in PBS and 1 nl injected into the duct of Cuvier. Larvae were transferred to fresh E3 to recover from anaesthetic. Any zebrafish injured by the needle/micro-injection, or where infection was not visually confirmed with the presence of phenol red, were removed from the procedure. Zebrafish were maintained at 28˚C.

## Microscopy of infected zebrafish

Larvae were anaesthetized 0.168 mg/mL tricaine in E3 and mounted in 0.8% low melting agarose onto glass bottom microwell dishes (MatTek P35G-1.5-14C). For low *C. neoformans* dose infection time points, confocal imaging was completed on a Zeiss LSM700 AxioObserver, with an EC Plan-Neofluar 10x/0.30 M27 objective. Three biological repeats contained 7, 10 and 12 infected zebrafish. Larvae were imaged in three positions to cover the entire larvae (head, trunk and tail) at 2 hpi, and at subsequent 24 hour intervals. After each imaging session, larvae were recovered into fresh E3 and returned to a 96-well plate.

A custom-build wide-field microscope was used for imaging blood vessel integrity in transgenic zebrafish after infection with *C. neoformans*. Nikon Ti-E with a CFI Plan Apochromat λ 10X, N.A.0.45 objective lens, a custom built 500 μm Piezo Z-stage (Mad City Labs, Madison, WI, USA) and using Intensilight fluorescent illumination with ET/sputtered series fluorescent filters 49002 and 49008 (Chroma, Bellow Falls, VT, USA). Images were captured with Neo sCMOS, 2560 × 2160 Format, 16.6 mm x 14.0 mm Sensor Size, 6.5 μm pixel size camera (Andor, Belfast, UK) and NIS-Elements (Nikon, Richmond, UK). Settings for *Tg(kdrl: mCherry)* and *TgBAC(ve-cad:GALFF)* crossed to *Tg(10xUAS:Teal)^{uq13bh}* GFP, filter 49002, 50 ms exposure, gain 4; mCherry, filter 49008, 50 ms exposure, gain 4. Settings for the GFP fluorescent beads were altered for GFP alone, filter 49002, 0.5 ms exposure, gain 4. In all cases a 50 μm z-stack section was imaged with 5 μm slices. Larvae were imaged at 2 hpi, and at subsequent 24 hour intervals. After each imaging session, larvae were recovered into fresh E3 and returned to a 96-well plate.

Co-injection of 40KDa FITC dextran with cryptococcal cells for imaging of brain vasculature was completed on 3 dpf immediately after dextran injection, using a Ziess Z1 light sheet obtained using Zen software. A W-Plan-apochromat 20x/1. UV-Vis lense was used to obtain z-stack images using the 488nm and 561nm lasers and a LP560 dichroic beam splitter.

## Time-lapse microscopy of infected zebrafish

For time-lapse imaging of *C.neoformans* low dose infection, larvae were anaesthetised and mounted as described above, with the addition of E3 containing 0.168 mg/mL tricaine overlaid on top of the mounted *nacre* larvae. Images were captured on the custom-build wide-field

microscope (as above), with CFI Plan Apochromat λ 10X, N.A.0.45 objective lens, using the settings; GFP filter 49002, 50 ms exposure, gain 4; mCherry filter 49008, 50 ms exposure, gain 4. Images were acquired with no added delay (~0.6 second intervals) for 1 hour, starting <2mins after infection.

### FRET microscopy and analysis

Larvae from the FRET tension sensor transgene line, *TgBAC(ve-cad:ve-cadTS)uq11bh* were infected with mCherry *C. neoformans* and mounted for imaging, as above. A spinning disc confocal microscope, (UltraVIEW VoX, Perkin Elmer, Cambridge, UK). A 40x oil lense (UplanSApo 40x oil (NA 1.3)) was used for imaging. TxRed, exitation 561nm with 525/640nm emission filter, CFP, exitation 440nm with 485nm emission filter, YFP, exitation 514nm with 587nm emission filter, and FRET exitation 440nm with 587nm emission filter were used as well as bright field images. All images were acquired using a Hamamatsu C9100-50 EM-CCD camera. Volocity (Perkin Elmer) software was used with this microscope for image capture. Analysis of images was completed using ImageJ software. The fluorescence signal intensity of the FRET, CFP and YFP channels was measured at each side of a vessel. This was completed at the location of a cryptococcal mass, or if no mass was present the middle of the vessel was measured. The FRET signal was then divided by the YPF signal, and an average was taken per vessel.

### Image analysis

The term cryptococcal mass has been used to describe the cryptococcal cells as well as associated capsule which are observed in blood vessels. Image analysis performed to measure the size of cryptococcal masses, and blood vessel width was completed using NIS elements. Fluorescence intensity of GFP and mCherry *C. neoformans* for low infection analysis was calculated using ImageJ software.

### Statistical analysis

Statistical analysis was performed as described in the results and figure legends. We used Graph Pad Prism 6–8 for statistical tests and plots.

## Supporting information

**S1 Fig. Injected ratio and number does not determine uncontrolled infection.** Infection of AB wild-type larvae with 5:1 ratio of GFP:mCherry KN99 *C. neoformans*, actual number of cryptococcal cells, both GFP and mCherry KN99 in 25 cfu injected grouped by majority colour outcome. Each bar represents an individual fish.
(TIF)

**S2 Fig. *Δgsc* does not affect blood vessel widening or frequency of trapping.** A-D: Infection of KDRL mCherry blood marker transgenic line with 1000 cfu *Δgsc* or its parental strain *C. neoformans* A Blocked vessels (% of all inter-segmental vessels) at 1 dpi (n = 2, +/- SD, Kruskal-Wallis test) B Blocked vessels (% of all inter-segmental vessels) at 3 dpi (n = 2, +/- SD, Kruskal-Wallis test) C Vessel width with or without *C. neoformans* at 1 dpi (n = 2, +/- SD, ns = not significant, ****p<0.0001, Kruskal-Wallis test) D Vessel width with or without *C. neoformans* at 3 dpi (n = 2, +/- SD, ns = not significant, ****p<0.0001, Kruskal-Wallis test)
(TIF)

**S3 Fig. *Δsmt* does not affect blood vessel widening or frequency of trapping.** A-D: Infection of KDRL mCherry blood marker transgenic line with 1000 cfu *Δsmt* or its parental strain *C. neoformans* A Blocked vessels (% of all inter-segmental vessels) at 1 dpi (n = 2, +/- SD, Kruskal-Wallis test) B Blocked vessels (% of all inter-segmental vessels) at 3 dpi (n = 2, +/- SD, Kruskal-Wallis test) C Vessel width with or without *C. neoformans* at 1 dpi (n = 2, +/- SD, ns = not significant, ****p<0.0001, Kruskal-Wallis test) D Vessel width with or without *C. neoformans* at 3 dpi (n = 2, +/- SD, ns = not significant, ****p<0.0001, Kruskal-Wallis test)
(TIF)

**S4 Fig. *Δsld8* does not affect blood vessel widening or frequency of trapping.** A-D: Infection of KDRL mCherry blood marker transgenic line with 1000 cfu *Δsld8* or its parental strain *C. neoformans* A Blocked vessels (% of all inter-segmental vessels) at 1 dpi (n = 2, +/- SD, Kruskal-Wallis test) B Blocked vessels (% of all inter-segmental vessels) at 3 dpi (n = 2, +/- SD, *p<0.05, Kruskal-Wallis test) C Vessel width with or without *C. neoformans* at 1 dpi (n = 2, +/- SD, ns = not significant, ****p<0.0001, Kruskal-Wallis test) D Vessel width with or without *C. neoformans* at 3 dpi (n = 2, +/- SD, ns = not significant, ****p<0.0001, Kruskal-Wallis test)
(TIF)

**S5 Fig. Phagocyte recruitment to cryptococcal masses within inter-segmental vessels.** A-C: Infection of *Tg(mpeg1:mCherry.CAAX)sh378* stably crossed to *Tg(mpx:eGFP)i114* larvae at 2 dpf with 1000 cfu KN99 *C. neoformans* imaged at 3 dpi A Example image of cryptococcal mass within inter-segmental vessel of larvae with no phagocyte recruitment B Example image of cryptococcal mass within inter-segmental vessel of larvae with both macrophage and neutrophil recruitment C Example image of cryptococcal mass within inter-segmental vessel of larvae with macrophage recruitment, also showing typical phagocytosis of just part of the cryptococcal mass.
(TIF)

**S1 Video. Cryptoccocal cell bud growth and separation in cryptococcal mass.** Time lapse transmitted light video (frames captured every 5 minutes) of cryptococcal cell proliferation within intersegmental vessel of zebrafish larva 3 dpf.
(AVI)

**S1 Data. Data values for figure graphs.**
(XLSX)

## Acknowledgments

We thank Timothy Chico (University of Sheffield, UK) and Robert Wilkinson (University of Sheffield, UK) for help and advice on vascular biology and Mike Tomlinson (University of Birmingham, UK) for help and advice on ADAM regulation of VE-Cadherin. We thank Arturo Casadevall (Johns Hopkins University, Maryland, USA) for providing the 18B7 antibody and Maurizio Del Poeta (Stony Brook Univeristy, New York, USA) for providing *Cryptococcus* ceramide pathway mutants. We thank aquarium staff at the Bateson Centre (Sheffield) and the IMCB (Singapore) for zebrafish husbandry.

## Author Contributions

**Conceptualization:** Josie F. Gibson, Philip W. Ingham, Stephen A. Renshaw, Simon A. Johnston.

**Data curation:** Josie F. Gibson.

**Formal analysis:** Josie F. Gibson, Aleksandra Bojarczuk, Alfred Alinafe Kamuyango, Richard Hotham.

**Funding acquisition:** Philip W. Ingham, Stephen A. Renshaw, Simon A. Johnston.

**Investigation:** Josie F. Gibson, Aleksandra Bojarczuk, Robert J. Evans, Richard Hotham.

**Methodology:** Josie F. Gibson, Simon A. Johnston.

**Project administration:** Josie F. Gibson, Simon A. Johnston.

**Resources:** Anne K. Lagendijk, Benjamin M. Hogan.

**Supervision:** Philip W. Ingham, Stephen A. Renshaw, Simon A. Johnston.

**Writing – original draft:** Josie F. Gibson, Philip W. Ingham, Stephen A. Renshaw, Simon A. Johnston.

**Writing – review & editing:** Josie F. Gibson, Aleksandra Bojarczuk, Robert J. Evans, Anne K. Lagendijk, Philip W. Ingham, Stephen A. Renshaw, Simon A. Johnston.

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
