## [Decision Letter · Decision Letter 0]

6 Jan 2022

Dear Dr. Johnston,

Thank you very much for submitting your manuscript "Blood vessel occlusion by Cryptococcus neoformans is a mechanism for haemorrhagic dissemination of infection" for consideration at PLOS Pathogens. Your manuscript was reviewed by members of the editorial board and by several independent reviewers. Based on the reviews, we are likely to accept this manuscript for publication, providing that you modify the manuscript according to the review recommendations.

Sincerely,

Xiaorong Lin, Ph.D.

Section Editor

PLOS Pathogens

Xiaorong Lin

Section Editor

PLOS Pathogens

Kasturi Haldar

Editor-in-Chief

PLOS Pathogens

orcid.org/0000-0001-5065-158X

Michael Malim

Editor-in-Chief

PLOS Pathogens

orcid.org/0000-0002-7699-2064

Reviewer Comments (if any, and for reference):

Reviewer's Responses to Questions

**Part I - Summary**

Reviewer #1: I was reviewer 1 in the previous submission: "Gibson and Bojarczuk et al present a very interesting manuscript describing how intravascular growth of Cryptococcus neoformans (Cn) correlates with the appearance of endothelial damage and preceding Cn invasion out of the vasculature. They make excellent use of dual fluorescent Cn to correlate individual instances of intravascular growth with localised pathologies. The second half of the manuscript focusses nicely on the Cnendothelial interaction and drills down into VE-cadherin junctional stiffness as a mechanistic determinant of tissue invasion"

Reviewer #2: The revised manuscript from Gibson et al. has added new data from dextran permeability experiments and immune cell localization that provide further evidence for the major findings of this manuscript. They have also expanded the text in a number of important ways that further contextualize their work. Together, these new additions to the manuscript have provided further support for their findings and their importance within pathogenesis. This work is well done and will be of broad interest within the pathogenesis community.

**Part II – Major Issues: Key Experiments Required for Acceptance**

Reviewer #1: The authors have addressed all of my concerns with either new experimental data, reinterpretation of existing datasets, or acceptable technical reasons why the experiments are not possible.

Reviewer #2: (No Response)

**Part III – Minor Issues: Editorial and Data Presentation Modifications**

Reviewer #1: None.

Reviewer #2: Line 224-225 – ‘infected blood vessels were larger relative to blood vessels in the same location in control animals’ this is a little confusing as the vessels/endothelial cells themselves aren't directly infected. Maybe something like 'vessels containing cryptococcus cells' would be more clear.

Line 213-225 - From the text, the distinctions from the results from experiments in Figure 4C and 4 G, H are not clear – expanding a bit to explain the distinctions here might be useful.

Line 304-310 - While I appreciate the efforts you have made to soften the language around endothelial junctions, the passage still gives the sense that the VE-cadherin/Gal4;UAS/Teal transgenic is providing some sort of readout of vascular integrity that was missed with the previous flk1 transgenics. However, since both of these reporters ultimately end up using an endothelial specific promoter to drive a cytoplasmic fluorescent protein, it doesn’t seem like this line is offering much new information on cell-cell junctions, particularly since VE-cadherin is also regulated in a number of ways not captured by this reporter (e.g. localization, posttranslational modification etc.) and even if its expression is regulated, with the long half-life of FPs in the cell, changes in VE-cadherin transcription may not even be visible within the time frame of these experiments. I think it’d be best to focus on intact and disrupted vessels alone and leave out any discussion of junctional integrity.

Line 309-310 – It would be useful to further explain the criteria for intact vs. disrupted vessels.

Line 758 – ‘Kymograph’ – this isn’t a Kymograph but rather a graph of width over time at a single point.

Line 802-803 the legend describes ‘Intact endothelial junctions’ – but as mentioned in the previous review, the transcriptional reporter doesn’t really capture the status of the junctions themselves.

Line 703-704 ‘1000/25 cfu of a 5:1 ratio’ what does 1000/25 cfu mean in this context?

Line 706 – ‘1000 cfu ratio of KN99’ missing a ratio.

Fig. 3 – There are some overlayed word outlines (DPI etc.) that appear in panels B and D.

Fig. 4D and E – Perhaps width of cryptococcus masses rather than ‘C. neoformans mass width’

PLOS authors have the option to publish the peer review history of their article (what does this mean?). If published, this will include your full peer review and any attached files.

Reviewer #1: **Yes: **Stefan Oehlers

Reviewer #2: No

Figure Files:

Data Requirements:

Reproducibility:

References:

---

## [Editor Report · Decision Letter 1]

21 Feb 2022

Dear Dr. Johnston,

We are pleased to inform you that your manuscript 'Blood vessel occlusion by Cryptococcus neoformans is a mechanism for haemorrhagic dissemination of infection' has been provisionally accepted for publication in PLOS Pathogens.

Best regards,

Xiaorong Lin, Ph.D.

Section Editor

PLOS Pathogens

Xiaorong Lin

Section Editor

PLOS Pathogens

Kasturi Haldar

Editor-in-Chief

PLOS Pathogens

orcid.org/0000-0001-5065-158X

Michael Malim

Editor-in-Chief

PLOS Pathogens

orcid.org/0000-0002-7699-2064
---

## [Editor Report · Acceptance letter]

18 Mar 2022

Dear Dr. Johnston,

We are delighted to inform you that your manuscript, "Blood vessel occlusion by *Cryptococcus neoformans* is a mechanism for haemorrhagic dissemination of infection," has been formally accepted for publication in PLOS Pathogens.

Best regards,

Kasturi Haldar

Editor-in-Chief

PLOS Pathogens

orcid.org/0000-0001-5065-158X

Michael Malim

Editor-in-Chief

PLOS Pathogens

orcid.org/0000-0002-7699-2064